# OpenReview forum: "Exposing Hidden Biases in Text-to-Image Models via Automated Prompt Search"
_ICML.cc/2026/Conference — ICML 2026 regular_

### Official Review · Reviewer_3GWL · 2026-03-03

**Soundness:** 3
**Presentation:** 3
**Significance:** 2
**Originality:** 2
**Overall Recommendation:** 5
**Confidence:** 4

**Summary:**

The authors investigate the problem of bias mitigation in text to image (TTI) diffusion models, especially when bias is implicitly triggered by contextual, linguistic or stylistic cues in prompts. Existing strategies for curating prompt datasets for debiasing rely on manual or LLM curation—which yields realistic test cases but does not cover all vulnerable corner cases; or, on gradient-based prompt optimisation, which discovers high bias regions but may produce incomprehensible text. The authors introduce Bias-Guided Prompt Search (BGPS), which automatically generates prompts (from LLMs) to maximise a joint objective of two terms: a bias score, to induce target biases in synthesised images; a language prior, to ensure semantic coherence of prompts sampled. The authors perform large-scale experiments on Stable Diffusion 1.5 and its debiased counterparts (finetuned with LoRA, plus with Difflens test-time debiasing); also, with SD 2.1, SDXL, DeepFloyd IF; they compare BGPS with other debiasing strategies: manual prompt curation, LLM prompt generation and the gradient-based, PEZ. Three LLM families are considered: Mistral, Qwen 3, Llama 3.2. Evaluating gender and race attributes, the authors detect residual demographic bias even in debiased models through BGPS in the form of extreme output demographic skew. In cases of baseline occupational bias, BGPS is reportedly able to further amplify biases whilst outputting coherent prompts. The authors provide a detailed breakdown of context-dependent bias amplification, where they ground the source of DM bias not to the occupation token, rather to stylistic or contextual modifiers. The authors decompose BGPS-generated prompts into word components – injected, deleted, dampened, augmented words – and argue that it is able to discover novel words that trigger biased attribute correlations.

**Compliance With Llm Reviewing Policy:**

Affirmed.

**Final Justification:**

The rebuttal meaningfully strengthens the paper by adding preliminary evidence that BGPS-generated prompts can improve debiasing itself, not just evaluation. The authors provide additional, helpful intuition about how/why residual bias may persist, e.g., through entangled contextual and stylistic features. My main concerns are sufficiently addressed, and I will raise my score from "weak accept" to "accept".

**Key Questions For Authors:**

- **Q1.** Have the authors considered using BGPS generated prompts for debiasing of diffusion models? E.g., with Difflens at test-time or LoRA finetuning? Does BGPS generate helpful data not only for evaluation but also bias mitigation?
- **Q2.** Why do the authors think that debiased models still struggle with contextual or stylistic modifier induced biases? Is this to do with a lack of factorisation and content-style disentanglement in the DMs' internal representations, or something else? Do the authors foresee any way of adding structure to BGPS's prompt generation procedures to more systematically assess and mitigate bias?
- **Comment 3.** The authors may want to consult metrics (including stylistic and behavioural ones) and methodologies utilised in the PRISM Alignment Dataset paper [1].

**References.**
[1] Kirk, Hannah Rose, et al. "The prism alignment dataset: What participatory, representative and individualised human feedback reveals about the subjective and multicultural alignment of large language models." Advances in Neural Information Processing Systems 37 (2024): 105236-105344.

**Limitations:**

Yes.

**Strengths And Weaknesses:**

### 1. Significance.
- **W1.1 – Extensions.** The reviewer believes that trying out BGPS generated prompts for active debiasing (beyond evaluation) would have greatly enriched the work, and substantiated claims that BGPS discovered novel, undocumented biases hidden in internal DM representations. Please refer to Q1 below.
- **W1.2 – Types of bias.** The reviewer recognises that it is very difficult to operationalise some notions of bias and fairness in machine learning. In this work, the authors focus on equalising over outcomes (perfect parity between attributes in the output distribution represents the golden, unbiased standard) but this may not be reflective of real world datasets and statistics or even use cases. A productive next step for BGPS may be to differentiate between benign and malignant forms of bias, where output skew is present in hateful or prejudiced contexts. One example is as follows: it is relatively benign to generate more male nurses than female ones, but it would be quite improper to generate more thieves of a particular race than other ones, since the occupation “thief” carries a strong negative connotation.

### 2. Significance.
- **S2.1 – Experimentation.** This is an experimental work on the bias evaluation of TTI diffusion models. Evaluation is comprehensive and conducted on relevant, large-scale models. Key modules of the proposed BGPS approach are ablated, e.g. choice of LLM for text generation (Mistral-7B-v0.2, Qwen 3, Llama 3.2), lambda parameter, and the type(s) of demographic bias. This gives confidence that their method is general purpose and broadly effective.
- **S2.2 – Comparisons.** The authors have made considerable effort to benchmark the BGPS approach against existing debiasing techniques of manual prompt curation and the gradient-based PEZ technique. They demonstrate statistically significant improvements in output distributional parity and prompt legibility that hold across experimental settings.
- **W2.3 – Qualitative analysis of biases.** The authors attempt to break down biases in categories of occupation and non-occupational, context-dependent biases (e.g. object, activity, context, place) but perhaps this decomposed analysis could have been even more systematic, such as with more granular categorical annotations and further stylistic analyses. Please refer to Comment 3 below (the PRISM paper).
### 3. Originality.
- **S3.1 – Text to image.** While there is significant prior work on bias assessment and mitigation in vision language models, the reviewer recognises that this work novelly introduces the BGPS framework to automatically search and sample comprehensible, bias maximising prompts. This is a non-trivial subtask and the technique proposed (which is VGD inspired) also represents a suitable and interesting use case for LLMs’ coherent language modelling abilities.
### 4. Presentation.
 - **S4.1 – Writing and clarity.** The writing is clear and the authors are able to well motivate their problem. They begin with a convincing discussion of why there exists a research gap (between gradient-based and handcrafted approaches) in generating prompt datasets for DM bias mitigation, and highlight a tension between coverage and comprehensibility in such prompts.
- **S4.2 – Results.** Claims that BGPS discovers and amplifies residual demographic biases are well supported by numerical reports and visualisations. The authors also clearly illustrate and ground which words within prompts trigger biases—the word clouds are a creative and effective way of reporting trigger word frequency and association—which does well to argue that biases are often induced by more implicit correlations.

---

> ### Author Rebuttal · Authors · 2026-03-31
>
> Below we address **all the reviewer’s comments** in the order they were presented:
>
> **W1.1 – Extensions, Q1: Do BGPS prompts help debiasing?**
>
> We agree with the reviewer that using BGPS-generated prompts for bias mitigation is a promising and important direction. In fact, we highlight this possibility in the related work section, where we discuss how automatically discovered prompts could complement existing debiasing pipelines by expanding the coverage of bias-triggering inputs.
>
> A full study of how to optimally integrate BGPS into different mitigation strategies (training protocols, scaling, interaction effects) is beyond our current scope, but we conducted a preliminary experiment to assess the research direction's plausibility.
>
> Specifically, we further fine-tuned the text encoder of a LoRA-debiased SD 1.5 model, comparing training on BGPS-generated prompts ("FT + BGPS") against training on standard occupation prompts alone ("FT"). Results for male biasing are shown across three BGPS search intensities (λ).
>
> We observe **an additional reduction in male proportions** compared to the pre-trained debiased model, and **a resistance to male bias increase** from BGPS prompts, even though BGPS is explicitly designed to amplify biased generations. This suggests that BGPS prompts can serve as informative hard examples for improving debiasing, and supports their potential utility beyond evaluation.
>
> | λ | | FT(BGPS prompts) | FT |
> |---|---|---|---|
> | **0** | Male % | 0.45 ± 0.06 | 0.57 ± 0.06 |
> |       | PPL | 98 ± 29 | 98 ± 29 |
> | **10** | Male % | 0.41 ± 0.05 | 0.60 ± 0.05 |
> |        | PPL | 83 ± 16 | 75 ± 8 |
> | **100** | Male % | 0.47 ± 0.05 | 0.71 ± 0.05 |
> |         | PPL | 93 ± 15 | 98 ± 17 |
>
>
> **W1.2 – Types of bias**
>
> We agree with the reviewer that there is a need to go beyond the simple idea of bias as a skew of the underlying “true” distribution of some attributes, but also take into account cultural and historical factors that make certain types of biases much more dangerous than others.  Our use of distributional parity as the evaluation criterion follows the standards set in the TTI bias literature which enables direct comparison with prior work. We will include a discussion of this point in the limitations section of the paper.
>
> **W2.3 – Qualitative analysis of biases, Comment 3: PRISM paper**
>
> While PRISM focuses on LLM alignment rather than T2I bias, its main highlight is that subjective phenomena such as bias should be assessed along multiple dimensions (e.g., appropriateness, harmfulness).
>
> Our current work, following the existing TTI bias evaluation and mitigation literature, evaluates bias in terms of distributional skew. An important extension, inspired by PRISM, could be to incorporate a richer human-centered evaluation of generated outputs, including annotating different types of bias (e.g., benign vs harmful) and analyzing how judgments vary across annotators. We also see value in extending our prompt decomposition with more fine-grained, structured categories of context and style.
>
> We will incorporate a discussion of these connections in the revision and consider them as directions for future work.
>
>
> **Q2 - Why do debiased models struggle?**
>
> We thank the reviewer for this insightful question. Our results suggest that current debiasing methods do not fully remove demographic associations carried by contextual and stylistic modifiers. A plausible explanation is that these models do not cleanly disentangle demographic attributes from other semantic and stylistic factors in their internal representations. As a result, modifiers such as scene, activity, object, or style descriptors can still activate correlated latent directions, even when explicit prompt forms have been debiased. This is consistent with our qualitative analysis, where BGPS often finds that the bias is not driven by the occupation token alone, but by contextual additions around it.
>
> To help this disentanglement, fine-tuning-based methods could explore training with a more varied set of prompts that explicitly include potentially bias-amplifying contextual and stylistic modifiers, to encourage the model to learn more disentangled representations. This is supported by the results showing BGPS prompts improve debiasing performance shown above.
>
> For methods based on activation steering, a better selection of features such as the SAE decomposition used in Difflens would be needed to identify and intervene on the relevant latent directions associated with bias, while preserving other semantic factors.

---

> > ### Author Rebuttal · Reviewer_3GWL · 2026-04-04
> >
> > The rebuttal meaningfully strengthens the paper by adding preliminary evidence that BGPS-generated prompts can improve debiasing itself, not just evaluation. The authors provide additional,  helpful intuition about how/why residual bias may persist, e.g., through entangled contextual and stylistic features.My main concerns are sufficiently addressed, and I will raise my score from "weak accept" to "accept".

---

### Official Review · Reviewer_a4d5 · 2026-03-10

**Soundness:** 3
**Presentation:** 3
**Significance:** 3
**Originality:** 3
**Overall Recommendation:** 5
**Confidence:** 3

**Summary:**

This work introduces Bias-Guided Prompt Search (BGPS), a method for discovering prompts that lead to biased generations in text-to-image diffusion models. The framework amplifies biases found in the internal representations of diffusion models while utilizing LLMs to ensure the resulting text remains legible and interpretable. The method successfully uncovers natural phrases and less obvious prompts that trigger biased generations, even in models that have already undergone debiasing.

**Compliance With Llm Reviewing Policy:**

Affirmed.

**Final Justification:**

My original recommendation to Accept (5) this paper stays unchanged. This is a technically solid paper with novel and important findings.

**Key Questions For Authors:**

* Regarding the DL baseline: How do you interpret the baseline results where manually curated prompts yield a 31% male? Does this indicate that the DL method inherently overcorrects?

**Limitations:**

yes

**Strengths And Weaknesses:**

## Strengths
* The experimental setup is sound and demonstrates effectiveness across LLMs of varying sizes.
* The paper is clearly written and logically structured.
* The method successfully uncovers novel, context-dependent biases triggered by natural phrases and subtle linguistic modifiers.
* The paper provides compelling evidence that biases are not fully mitigated by current debiasing approaches, effectively highlighting a critical limitation of existing techniques.

## Weaknesses

* The evaluation using the Difflens (DL) debiasing method reveals an unusual shift in bias rather than true mitigation. For example, in the male-biasing experiment, the DL model generates male depictions only 31% of the time for manually curated prompts (where it should generate them in about 50% of cases). This suggests the baseline debiasing method overcorrected, producing a model heavily biased in the opposite direction.

---

> ### Author Rebuttal · Authors · 2026-03-31
>
> Below we answer to the only concern by the reviewer:
>
> **1) Does Difflens overcorrect?**
>
> Difflens, like many debiasing methods based on activation steering, modifies internal representations associated with specific attributes by scaling them during generation. In particular, it applies a linear intervention controlled by a hyperparameter (editing strength), which must be tuned to achieve equal distribution of target attributes.
>
> In our experiments, we used the editing strength recommended by the original Difflens paper for gender debiasing, while evaluating on manually curated prompts from the fine-tuning-based baseline. Under this setup, Difflens produces a noticeable shift toward the opposite gender (e.g., 31% male), which suggests overcorrection.
>
> We interpret this as a limitation of activation steering approaches more broadly: a fixed intervention strength may not transfer across different prompt distributions, and can lead to under- or over-correction depending on the evaluation setting.
>
> Importantly, this behavior **does not affect our main conclusions**: BGPS is evaluated comparatively across all methods under the same settings, and still consistently identifies prompts that expose residual bias (or shifts in bias) in the debiased models.

---

> > ### Author Rebuttal · Reviewer_a4d5 · 2026-04-02
> >
> > Thank you for this clarification, I maintain my original positive score.

---

### Official Review · Reviewer_6v8n · 2026-03-13

**Soundness:** 3
**Presentation:** 2
**Significance:** 2
**Originality:** 2
**Overall Recommendation:** 4
**Confidence:** 3

**Summary:**

The authors propose Bias-Guided Prompt Search (BGPS), a framework for generating prompts that are effective at exposing biases in text-to-image models while remaining interpretable. They steer the prompt generation process by scoring prompts that lead to images whose UNet activations are classified by an auxiliary classifier as reflecting a target bias. They evaluate their approach on debiased models and show that even these models remain vulnerable to hidden biases. They also assess the interpretability of the discovered prompts using perplexity as a measure of prompt naturalness

**Compliance With Llm Reviewing Policy:**

Affirmed.

**Final Justification:**

The response addressed almost all of my concerns, and I have increased my score accordingly.

**Key Questions For Authors:**

The method relies on classification based on UNet activations. How would this approach be applied to other text-to-image generative model architectures where the internal structure may differ or such intermediate activations may not be as accessible?

Could the authors provide a clearer runtime or efficiency analysis to clarify the practical computational cost of the method?

In Table 1, how should the improvement of BGPS over the LLM (biased) baseline be interpreted when the percentage of explicitly gendered prompts also increases? Could the authors provide a controlled comparison, for example after filtering out prompts that contain explicit gender words?

**Limitations:**

yes

**Strengths And Weaknesses:**

Strengths:

1. The paper analyses an important problem in text-to-image generative models.

2. The main idea of the paper is well defined and understandable to balance the effectiveness and interpretability of the prompts.

3. The paper shows debiased models still are vulnarable which is interesting.

Weaknesses

1. The paper presents itself as exposing hidden biases in text-to-image models in general, but its primary evaluation is centered on Stable Diffusion 1.5 and two debiased variants of SD1.5. Although they also conduct experiments on SD 2.1, SDXL, and DeepFloyd IF, the empirical coverage is still relatively limited for such a broad claim. Stronger support would require more experiments on newer models and also on other families such as Flux.

2. In several cases, the improvement of BGPS (𝜆=100) over LLM (biased) should be interpreted together with the increase in the percentage of explicitly gendered prompts. This makes it unclear how much of the gain comes from uncovering more subtle hidden bias versus allowing more explicit gender leakage into the prompt.

3. Based on the general claim of the paper, it is unclear how it can be applied to other architectures in text-to-image generative models. The approach depends on classifiers trained on UNet middle-layer activations, it is not clear how easily it can be extended to other text-to-image architectures.

4. Although the approach avoids gradient-based optimization through the full diffusion process, it still scores each prompt candidate after running T denoising steps for K random latents within a beam-search procedure. Given the beam size and expansion factors used in the paper, this still appears to be a relatively expensive search process. A more explicit runtime or efficiency analysis would help clarify the practical cost of the method.

---

> ### Author Rebuttal · Authors · 2026-03-31
>
> Below we address **all the reviewer’s comments**:
>
> **1)Different Architectures**
>
> Our paper focused on the widely used UNet-based diffusion architecture, but BGPS is a general-purpose algorithm applicable to any TTI model with access to its internal activations. Following the reviewer's suggestion, **we extended our method to the Flux model, which is based on the Diffusion Transformer (DiT) architecture**. Specifically, we trained a gender classifier on Flux's internal activations from the middle transformer layer residual stream, and found that BGPS successfully exposes biases in this architecture as well, confirming the generality of our approach. Because the choice of $\lambda$ is model-dependent (see Appendix F.3), after we saw performance decreasing for $\lambda=100$, we also added the $\lambda=50$ intermediate value as a better alternative. We present the results below and will include the full experiment in the revised version, along with details on adapting BGPS to DiT architectures.
>
> | λ| Male % | Male PPL| Female %| Female PPL|
> |-----------|----------------|-----------------|------------------|-------------------|
> | 0 (LLM only) | 0.55 ± 0.07 | 78 ± 8 | 0.42 ± 0.07 | 78 ± 8 |
> | 5 | 0.68 ± 0.07 | 121 ± 17 | 0.40 ± 0.08 | 99 ± 15 |
> | 10 | 0.62 ± 0.08 | 135 ± 21 | 0.53 ± 0.08 | 103 ± 16 |
> | 50 | 0.66 ± 0.10 | 122 ± 24 | 0.53 ± 0.10 | 146 ± 29 |
> | 100 | 0.61 ± 0.08 | 137 ± 23 | 0.46 ± 0.08 | 115 ± 22 |
>
> **2)Gendered Prompts:**
>
> Please refer to our answer to reviewer **vsYR** regarding explicitly gendered prompts:
>
> Below are Mistral 7B results with gendered prompts removed for the Base model. For λ=100, **mean male proportion drops from 0.92 to 0.91 and female from 0.67 to 0.66**, confirming BGPS's strength is driven by implicit rather than explicit gender biases. FT and DL models (omitted for space) also show negligible reductions after filtering, they will be included in the revised paper.
>
> Gendered prompts were not filtered-out in the main paper to show that: 1) PEZ produces virtually no gender-neutral prompts, making filtered comparison impossible, 2) high lambda causes the LLM to produce increasingly gendered prompts.
>
> We will clarify this in the main paper and include filtered results in the Appendix.
>
> | Method | Male % Base ↑ | Female % Base ↑ | PPL Base ↓ | Gendered % M/F |
> |-----|-------------|--------------|------------|-----------|
> | Curated | 0.53 ± 0.02 | 0.47 ± 0.02 | 96 ± 3 | 0/0 |
> | LLM | 0.69 ± 0.07 | 0.27 ± 0.06 | 72 ± 13 | 1/1 |
> | LLM (biased) | 0.67 ± 0.06 | 0.33 ± 0.06 | 60 ± 5 | 0/0 |
> | BGPS (λ=10) | 0.76 ± 0.06 | 0.43 ± 0.05 | 53 ± 5 | 2/0 |
> | BGPS (λ=100) | 0.91 ± 0.03 | 0.66 ± 0.06 | 122 ± 35 | 17/16 |
>
> **3)Method runtime/efficiency**
>
> BGPS uses information from the text-to-image model activations to guide the generation process, which requires evaluating at least one diffusion timestep per decoding step. The computational cost of BGPS (compared to LLM-only decoding) is dominated by K × B × E single-step UNet evaluations per decoding step, totaling K × B × E × L evaluations for a prompt of length L. With our default settings this yields ~20,000 UNet forward passes per prompt. On a H100 GPU this takes about ~2.4 minutes per prompt. Note that this cost is irrespective of T’, as according to the BGPS algorithm, we only evaluate the first T’ diffusion timesteps for K latents once, which for T’=25 is 250 evaluations, which is negligible.
>
> As this cost scales linearly with each factor, it can be reduced significantly by halving K from 10 to 5 (~1.2 min/prompt) with minimal impact on performance (see Appendix H.1). Generating a complete evaluation set of 100 prompts requires ~2–4 GPU-hours, a one-time auditing cost.
>
> We will include the above runtime analysis in the main paper.

---

> > ### Author Rebuttal · Reviewer_6v8n · 2026-04-02
> >
> > Thank you to the authors for their rebuttal. The response addressed almost all of my concerns, and I have increased my score accordingly. The added DiT-based experiment is helpful and strengthens the paper. However, I still think the paper would benefit from more extensive DiT-based experiments to better support the broader claim of generality. I also note that the method still relies on training an auxiliary classifier on internal latent representations. As a result, its applicability to pixel-space text-to-image generators remains somewhat unclear, both in terms of how the method would be formulated and whether it would remain computationally efficient

---

> > > ### Author Response · Authors · 2026-04-06
> > >
> > > We thank the reviewer for acknowledging our response, and for the constructive feedback. Some additional clarifications answering the reviewer's remaining concerns:
> > >
> > > **Extra DiT Experiments**: To make our experiments on diffusion transformers more complete, we also apply BGPS on the Stable Diffusion 3 model family, specifically on SD 3.5 medium. The SD3 model family differs from Flux as it uses a Multi-Modal Diffusion Transformer (MMDiT) architecture, that processes the text and image sequences separately, only performing cross-attention on the sequences at every block. In contrast, in the Flux architecture, after 19 dual stream blocks that are similar to MMDiT, the two sequences are combined in a joint text-image sequence, processed by single-stream transformer blocks (it is these single stream activations that BGPS uses to train the attribute classifier). We find that an attribute classifier with ~99% accuracy can be trained only on the SD3 image stream, and used for BGPS for finding novel biased prompts.
> > >
> > > | λ| Male %| Male PPL| Female %| Female PPL|
> > > |-----------|-----------------|-----------------|-------------------|-------------------|
> > > | 0   | 0.51 ± 0.08 | 78 ± 8  | 0.31 ± 0.07 | 78 ± 8  |
> > > | 10  | 0.72 ± 0.07 | 107 ± 11 | 0.38 ± 0.07 | 86 ± 10  |
> > > | 100 | 0.72 ± 0.08 | 122 ± 17 | 0.42 ± 0.08 | 91 ± 10  |
> > >
> > >
> > > **Latent vs pixel-space models**: Our method can successfully be applied to diffusion models that operate *both in VAE latent space as well as pixel-space models*, as indicated by our experiments on DeepFloyd IF (see Appendix F.3), which is a cascaded TTI model based on a pixel-space diffusion model followed by two upsampling models. It is the most powerful and recent open-weight diffusion model working in pixel-space.
> > >
> > > Regarding the computational efficiency of our method on different input modalities, as mentioned in our computational complexity analysis above, our method's main bottleneck is the need for multiple forward evaluations of the diffusion model. This means that regardless of input representation (latent or pixel-space) the runtime is dependent on the model runtime. For DiT architectures, this is exacerbated by the transformer quadratic attention complexity over sequence length.
> > >
> > > For example, training the attribute classifier for the Stable Diffusion 1.5 model (less than 1B parameters) takes ~70 seconds per epoch on an Nvidia H100 GPU, for DeepFloyd IF (4.5 B parameters) takes ~150 seconds. For Stable Diffusion 3.5 Medium, which is about half the parameter count of DeepFloyd IF, it takes ~1500 seconds (x20) for each epoch.

---

### Official Review · Reviewer_vsYR · 2026-03-16

**Soundness:** 2
**Presentation:** 3
**Significance:** 2
**Originality:** 2
**Overall Recommendation:** 4
**Confidence:** 4

**Summary:**

This paper introduces a method, BGPS, to automatically generate prompts that maximize the presence of certain biases in the generated images (from text-to-image models). BGPS utilizes automated prompt search and an attribute classifier to search for prompts, especially implicit ones, i.e., no specific identities within prompts. Their experiment results on text-to-image models seem effective. From the results, the authors offer many insights regarding how implicit words (contextual modifiers and so on) are somehow associated with the biased generation, which should be further mitigated for the debiased models.

**Compliance With Llm Reviewing Policy:**

Affirmed.

**Final Justification:**

The authors conducted the required experiments and made the goal and evaluation consistent, which clarifies the motivation of the work. They also included aggregate results across different attributes rather than focusing on only one, and showed that the strong performance remains even after removing gendered prompts. Given the current evidence, I am inclined to accept this paper.

**Key Questions For Authors:**

My largest concern with this paper focuses on the threat model and the evaluation design.

First, although the authors did not explicitly write their threat model, the readers understand that their goal is to design a method to **search for prompts** that could lead to biased exposure. The problem is, according to the way authors motivate this idea, their method is expected to address two limitations of current bias evaluation: coverage and interpretability. In this case, the actual goal is to search for **a wide range** of **interpretable** prompts, and the two dimensions should **both** be evaluated in the following experiments. Otherwise, the reviewer sees inconsistencies between the study goal and the evaluation design.

Second, the evaluation section currently reports the probability of biased generation, PPL, and how many prompts are “gendered.” Two issues here: (1) I think any “gendered” prompts should be removed from the evaluation set before reporting the performance. This is because the images generated from the “gendered” prompts substantially increase the presence of this gender in images, therefore inflating the reporting metric; (2) GPT-2-generated PPL is only a weak proxy for quantifying how natural, readable, and interpretable the prompt is. More metrics such as fluency, coherence, and LLM evaluator can be combined for judgement.

Finally, the current main experiments only cover two attributes: gender and race, and the main result in Table 1 only reports “male-biased prompts.” This is insufficient to demonstrate the effectiveness of BGPS. Generally, it is more common to report an aggregated result showing that the method performs well across multiple groups/subsets, rather than in only one case.

Additionally, I didn’t see the ablation study regarding the optimal lambda selection. The readers are left unclear which lambda achieves the best trade-off between coverage and interpretability.

One thing to improve: I feel the contribution of introducing a method that creates prompts is much weaker than directly contributing a comprehensive prompt set produced by this method, since BGPS’s ultimate goal is to create prompts for model developers to mitigate biased generation. So the authors might want to take a step forward and make BGPS immediately practical in curating the implicit biased prompt dataset. This requires a more comprehensive evaluation that covers multiple attributes, e.g., gender, race, age groups, etc.

**To sum up:**

(1) What is the actual goal? Why does the evaluation not report “coverage” or “diversity” of the searched prompts?

(2) Why are “gendered” prompts included in the evaluation?

(3) What is the optimal lambda that can be readily adopted in practice?

**Strengths And Weaknesses:**

Strengths

- This paper investigates an interesting and important problem: bias exposure
- The methodology of automated prompt search is effective
- There are many interesting findings regarding the associations between specific words/phrases and biased generation

Weaknesses

- The threat model is not fully clear
- The evaluation is both insufficient and unfair

---

> ### Author Rebuttal · Authors · 2026-03-31
>
> We address **all the reviewer’s comments**, in the order they were presented:
>
> **1) Paper goal and coverage/diversity:**
>
> Our goal is dual: (1) develop an attack that exposes hidden biases in text-to-image models via interpretable prompts, (2) augment human-curated fairness benchmarks with less obvious bias-triggering cases.
>
> We will also clarify the threat model in the revision: BGPS operates as a grey-box auditing method for model developers and researchers to stress-test bias mitigation approaches using interpretable prompts.
>
> Diversity:
> We agree that coverage/diversity should be evaluated more explicitly. To address this, we conducted an additional analysis using lexical diversity (Distinct-1/2/3), lexical similarity (Self-BLEU), and semantic similarity (sentence embedding-based) metrics. BGPS produces prompts that are **at least as (and most often more) diverse as LLM-only prompts**, while achieving stronger bias exposure.
>
> | Method | Distinct-1 ↑ | Distinct-2 ↑ | Distinct-3 ↑ | Self-BLEU ↓ | Emb. Sim ↓ |
> |---|---|---|---|---|---|
> | LLM only | 0.38 | 0.62 | 0.79 | 0.35 ± 0.03 | 0.244 ± 0.002 |
> | BGPS (λ=10) | 0.38 | 0.63 | 0.80 | 0.32 ± 0.03 | 0.249 ± 0.002 |
> | BGPS (λ=100) | **0.43** | **0.71** | **0.87** | **0.22 ± 0.02** | **0.222 ± 0.002** |
>
> **2)Gendered prompts:**
>
> Below are Mistral 7B results with gendered prompts removed for the Base model. For λ=100, **mean male proportion only drops from 0.92 to 0.91 and female from 0.67 to 0.66**, confirming BGPS's strength is driven by implicit rather than explicit gender biases. FT and DL models (omitted for space) also show negligible reductions after filtering, they will be included in the revised paper.
>
> Gendered prompts were not filtered-out in the main paper to show that: 1) PEZ produces virtually no gender-neutral prompts, making filtered comparison impossible, 2) high lambda causes the LLM to produce increasingly gendered prompts.
>
> We will clarify this in the main paper and include filtered results in the Appendix.
>
> | Method | Male Base ↑ | Female Base ↑ | PPL Base ↓ | Gen.% M/F |
> |-----|-------------|--------------|------------|-----------|
> | Curated | 0.53 ± 0.02 | 0.47 ± 0.02 | 96 ± 3 | 0/0 |
> | LLM | 0.69 ± 0.07 | 0.27 ± 0.06 | 72 ± 13 | 1/1 |
> | LLM (biased) | 0.67 ± 0.06 | 0.33 ± 0.06 | 60 ± 5 | 0/0 |
> | BGPS (λ=10) | 0.76 ± 0.06 | 0.43 ± 0.05 | 53 ± 5 | 2/0 |
> | BGPS (λ=100) | 0.91 ± 0.03 | 0.66 ± 0.06 | 122 ± 35 | 17/16 |
>
> **3)Naturalness of prompts:**
>
> Perplexity is widely used to measure text fluency [1]. Following the reviewer's suggestions, we also use GPT-5 as a judge to score fluency, coherence, and plausibility. Results confirm BGPS generates reasonably natural prompts at moderate λ with a controllable interpretability–bias tradeoff.
>
> | Method        | Fluency | Coherence | Plausibility |
> |---------------|--------:|----------:|-------------:|
> | manual        | 4.24    | 4.62      | 3.80         |
> | llm           | 3.69    | 3.88      | 3.76         |
> | l=10          | 3.54    | 3.67      | 2.82         |
> | l=100         | 2.8     | 3.2       | 2.6          |
> | PEZ           | 2.64    | 1.74      | 1.74         |
>
> **4) Experiments on different attributes and aggregate metrics:**
>
> Overall, BGPS is evaluated across multiple attributes (gender, race, age), three LLMs, four T2I models, and diverse prompt settings, with most results referenced in the main text but moved to the appendix due to space constraints. We agree that an aggregate view would be helpful, and will include the following table summarizing all six attributes (female, male, white, black, young, old) for the main experiment, showing **consistent improvements vs all baselines**.
>
> |  | vs. LLM |  | vs. Manually Curated |  |
> |---|---|---|---|---|
> | **Method** | **ΔAttr ↑** | **ΔPPL ↓** | **ΔAttr ↑** | **ΔPPL ↓** |
> | PEZ | +0.10 | +2026 | +0.08 | +1988 |
> | BGPS (λ=10) | +0.04 | **−7** | +0.04 | **−48** |
> | BGPS (λ=100) | **+0.20** | +20 | **+0.22** | −10 |
>
> **5)Lambda selection:**
>
> The ablation on $\lambda$ is included in Appendix D (p.14) in the original submission, showing the biasing strength vs. naturalness tradeoff. This means that the optimal choice of lambda depends on perplexity tolerance and required biasing strength, while also being dependent on the TTI model used (see Appendix F.3).
>
> **6)Using BGPS to create curated biased prompt set:**
>
> We agree with the reviewer that a comprehensive prompt set produced by our method would be a valuable contribution to the community. We will upload a curated prompt set in the project repository, with prompts for all scenarios in the paper and all explicitly gendered or bad quality prompts filtered out.
>
> ---
> [1] Subham Sahoo et al. Simple and effective masked diffusion language models. Advances in Neural Information Processing Systems, 37:130136–130184, 2024.
> [2] Donghoon Kim et al. Visually guided decoding: Gradient-free hard prompt inversion with language models. In International Conference on Learning Representations (ICLR), 2025

---

> > ### Author Rebuttal · Reviewer_vsYR · 2026-04-02
> >
> > Thank the authors for the detailed clarification and additional experimental results. I would raise the score if the committed changes are included in the revision.

---

> > > ### Author Response · Authors · 2026-04-02
> > >
> > > We thank the reviewer for acknowledging our responses and for the constructive feedback throughout the review process. Following ICML rebuttal rules (https://icml.cc/Conferences/2026/PeerReviewFAQ), we are not able to update the original submission during the rebuttal period. We confirm that all committed changes will be incorporated in the revised manuscript, should the paper become accepted.

---

### Decision · Program_Chairs · 2026-04-30

**Decision:**

Accept (regular)

**Comment:**

This paper has received positive final recommendations, with two Weak Accept and two Accept recommendations.

The reviewers generally recognized the novelty of the proposed method, the importance of studying bias in Text-to-Image diffusion models, the effectiveness of the designed automated prompt search, supported by comparisons with benchmark methods and visualizations. Furthermore, the paper is recognized as well-structured, well-written and easy to read, and the provided experiments showing how debiasing model can still be vulnerable to hidden bias are considered interesting and compelling.

On the other hand, some weaknesses are initially identified, including the necessity to test the proposed approach on newer models, or on Flux, the specific sources of gains, applicability to other architectures rather than UNet, the necessity to include and test multiple sensitive attributes, as well as requests for extensions to preliminary experiments for actual debiasing and not only evaluation.

In the rebuttal, the authors provided new experiments and clarifications, and all the reviewers generally acknowledged the effectiveness of additional evidence and specifications provided, resulting in a final overall positive evaluation, given that the committed changes are included in the revision.

After careful evaluation of the initial reviews, the rebuttal and discussion with the authors, the AC recognizes the value of this work, and agrees on the identified strengths, therefore recommending acceptance of the paper.
The authors are encouraged to include the committed changes in the camera-ready version of this work.